# Detection of Movement-Related Brain Activity Associated with Hand and Tongue Movements from Single-Trial Around-Ear EEG

**DOI:** 10.3390/s24186004

**Published:** 2024-09-17

**Authors:** Dávid Gulyás, Mads Jochumsen

**Affiliations:** Department of Health Science and Technology, Aalborg University, 9260 Gistrup, Denmark; dgu@hst.aau.dk

**Keywords:** brain–computer interface, ear-EEG, movement intention, movement-related cortical potentials, sensorimotor rhythm, hand, tongue

## Abstract

Movement intentions of motor impaired individuals can be detected in laboratory settings via electroencephalography Brain–Computer Interfaces (EEG-BCIs) and used for motor rehabilitation and external system control. The real-world BCI use is limited by the costly, time-consuming, obtrusive, and uncomfortable setup of scalp EEG. Ear-EEG offers a faster, more convenient, and more aesthetic setup for recording EEG, but previous work using expensive amplifiers detected motor intentions at chance level. This study investigates the feasibility of a low-cost ear-EEG BCI for the detection of tongue and hand movements for rehabilitation and control purposes. In this study, ten able-bodied participants performed 100 right wrist extensions and 100 tongue-palate movements while three channels of EEG were recorded around the left ear. Offline movement vs. idle activity classification of ear-EEG was performed using temporal and spectral features classified with Random Forest, Support Vector Machine, K-Nearest Neighbours, and Linear Discriminant Analysis in three scenarios: Hand (rehabilitation purpose), hand (control purpose), and tongue (control purpose). The classification accuracies reached 70%, 73%, and 83%, respectively, which was significantly higher than chance level. These results suggest that a low-cost ear-EEG BCI can detect movement intentions for rehabilitation and control purposes. Future studies should include online BCI use with the intended user group in real-life settings.

## 1. Introduction

Motor impairments limit the individuals’ ability to interact with the world and communicate with others [1,2,3]. Different aids and technologies exist for alleviating the impairment, but the ability to control them depends on the severity of the impairment. For individuals with debilitating impairments brain–computer interfaces (BCIs) may be used [4]. BCI systems allow the user to control external devices such as wheelchairs, robotic manipulators, exoskeletons and communication aids which can be used to replace lost functions. Potential user groups for such technology are patients with, e.g., amyotrophic lateral sclerosis (ALS) and spinal cord injury. These individuals are able to produce control signals that can be extracted from the electrical brain activity and used for controlling the BCI [5,6] even in the locked-in state of ALS [7].

Some of the most common control signals for controlling BCIs are steady-state visual evoked potentials, P300 and motor imagery. The former two are generally associated with higher information transfer rates compared to motor imagery, and hence more sophisticated control of the external technology, but they depend on external stimuli such that the user is paced by the system contrary to motor imagery, which can be operated in a self-paced manner without the need for external stimuli. Motor imagery is elicited by an imagined movement that activates the areas in the brain associated with movement programming, which are also activated in association with executed and attempted movements [8]. Motor imagery or movement-related brain activity are also used in BCIs for restorative purposes where neuroplasticity is induced especially for stroke rehabilitation [9,10,11]. Neuroplasticity is induced when the brain activity from areas associated with movement preparation is paired with afferent inflow of somatosensory feedback [12,13,14,15].

Despite the potential of the technology to improve the user’s independence or rehabilitation outcome, current BCI use is mostly limited to the laboratory due to various factors. These include the complexity and cumbersomeness of the setup [16,17,18,19,20], bulkiness and price of the amplifiers [16,20,21,22], unstable performance due to incorrect mounting of the cap or poor signal-to-noise ratio. Moreover, the use of caps may be impractical due to gel in the hair and the feeling of physical discomfort when wearing them [16,18,19,20]. The aesthetics of the BCI also play a role in the adoption of the technology, as users are not willing to wear devices that identify them as patients in public or even their community [16,18,20,21]. Thus, there is a need for low-cost, lightweight, comfortable, and unobtrusive BCI systems that do not require gel-based electrodes in the hair.

In-ear or around-ear electroencephalography (EEG) are potential candidates, being compact, located outside hair and unobtrusive. While the use of ear-EEGs has been widely investigated in a number of contexts [23], motor-task ear-EEG BCI systems remain under-explored since the recording of motor-related activity often requires electrodes placed over the motor cortical areas (see Table 1); previous experiments have focused on hand movements [24,25,26,27], and motor execution [24,25,26,28] and only two studies attempted to detect movement intentions among idle epochs [24,25]. On the other hand, the variety of pre-processing and classification algorithms tested on ear-EEG motor-task data has been high. A filter-focused approach was taken by Kim et al. [27] who used common spatial patterns and a regularized linear discriminant analysis classifier on band-pass filtered data. The optimal frequency band for the bandpass filter was narrowed down from the default 5–35 Hz range in a five-fold cross-validation. While the results reported in Table 1 are above chance level, the use of a reference electrode away from the ear does not offer the practicality of ear-EEG-only setups, as such BCI system cannot be concealed inside, e.g., headphone-like devices. The authors compared their ear-EEG results with classification on 21-channel scalp EEG which achieved 92.4% accuracy on BCI competition and 91.6% on the authors’ dataset. A neural network approach was proposed by Wu et al. [26]. A dataset of 160 trials per subject (apart from one subject with only 80 trials) was used to train and evaluate EEGNet. Both within-subject and cross-subject training yielded near-chance level results, but when the neural network was fine-tuned on the subject being evaluated the accuracy improved (see Table 1). It is worth noting that both increasing the electrode distance from in-ear to near-ear setup and moving the reference from the contralateral ear to the Cz channel had a positive effect on the classification accuracy by 5% and 2%, respectively. Jochumsen et al. [25] constructed manual time-domain, spectral, and template features, that were used in Random Forest (RF) with 512 trees for movement intention detection. Ten-channel ear-EEG accuracy reached 56% and 60% on days 1 and 2, respectively, which did not significantly differ from random chance. The authors also collected nine-channel scalp-EEG, which could detect movement intention with an accuracy of 77% and 74% on days 1 and day 2, respectively. In a study by Schalk et al. [24], subjects were prompted to simultaneously close both their hands while the authors collected three-channel ear-EEG. For each channel, movement intention was detected from spectral EEG information between 0.3 and 40 Hz using linear discriminant analysis. The average movement intention detection accuracy for ear-EEG channels was 54%, and for C4 and Cz channels it was 69%. Two major limitations of this study include single-channel movement intention detection and a lack of noisy epoch removal. Kæseler et al. [28] ran an ear-EEG tongue movement detection study. The authors have not trained a classifier on ear EEG-only data, but they reported visible tongue-movement-related cortical potentials (MRCPs) on T7 and T8 electrodes. Detection of tongue movements from ear-EEGs is of particular interest, as the tongue’s representation in the primary motor cortex is located closer to the ears than the representation of the lower and upper extremities [29]. In the existing studies, data collection has been performed exclusively using laboratory-grade amplifiers [24,25,26,28], so the feasibility of using a light-weight low-cost amplifier is unclear.

**Table 1 sensors-24-06004-t001:** An overview of studies investigating the use of ear-EEGs for motor task BCIs. N designates the number of subjects (all subjects are able-bodied). Electrode locations are reported according to Figure 1. Reference electrode xOU refers to the contralateral OU electrode as seen in Figure 1. Reference electrodes that are limited to ear area are **bolded**. Classes are reported in the manner in which they were classified (A vs. B) using the following abbreviations: L (left hand), R (right hand), F (foot), I (idle, no movement). * = BCI Competition III, dataset IVa [30].

	N	Electrode Location (Number), Reference	Task: (B) Ballistic or (I) Isometric	Amplifier	Classes	Classifier	Highest Achieved Accuracy
[24]	15	Around-ear (3), **A2**	Palmar grasp (I)	g. HIamp, g.tec	L + R vs. I	LDA	54%
[28]	10	Near-ear (2), **A2**	Tongue-palate touch	g. HIamp, g.tec			
[25]	12	CEEGrid (10), **R8**	Palmar grasp (B)	Mobita, TMSi	R vs. I	RF	60%
[26]	6	In-ear (8), Cz	Palmar grasp (I)	SynAmps2	L vs. R	EEGNet	70%
		In-ear (8), **xOU**					68%
		Near-ear (6), Cz					75%
[27]	5	Near-ear (14), nose tip	Unspecified MI (I)	Unspecified	R vs. F	RLDA	72%
	5	Near-ear (8) *	Unspecified MI	Unspecified			68%

**Figure 1 sensors-24-06004-f001:**
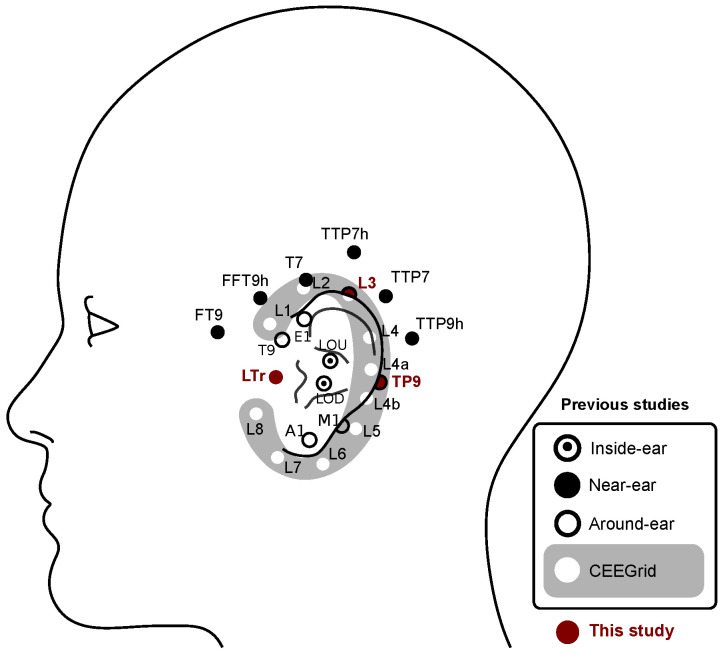
Overview of ear-EEG electrode locations used in the current and the previous motor-task studies. The two electrodes (LB and LF) are located too deep inside the ear canal to be visible and are omitted from the figure. CEEGrid electrodes are positioned behind the ear. Only left-hemisphere ear-EEG electrodes are shown here, but some studies collected right-hemisphere ear-EEG, too.

To facilitate the adoption of BCIs by the user, this study attempts to address the impracticality of in-hair EEG setup and the cost and bulkiness of standard EEG amplifiers. For that purpose, the detection of movement-related brain activity collected with a low-cost light-weight and few-channel ear-EEG BCI system was evaluated. It was evaluated in an offline manner for three use cases: (1) using movement-related brain activity associated with tongue movement (control applications), (2) using movement-related brain activity associated with hand movement (control applications), and (3) using movement-related brain activity associated with hand movement (rehabilitation applications). The difference between use cases 2 and 3 is that only information until the movement onset was used in use case 3 to adhere to the strict temporal timing of afferent inflow of the somatosensory feedback.

## 2. Materials and Methods

### 2.1. Participants

Ten able-bodied participants (five women, four men, one non-binary, age: 25.9 ± 3.7 years, nine right-handed and one left-handed, nine with prior BCI experience) consisting of university students and staff participated in this study. Prior to the experiment, all subjects provided their written informed consent. All procedures were approved by The North Denmark Region Committee on Health Research Ethics (approval number: N-20230015).

### 2.2. Data Collection

Continuous EEG was collected using a Cyton Biosensing Board (OpenBCI, Brooklyn, NY, USA) at 250 Hz using Neuroline 720 self-adhesive wet electrodes (AMBU A/S, Ballerup, Denmark). Three channels around the left ear were recorded; in front of the left tragus—Front (LTr), above the left upper helix—Over (L3), and on the left mastoid Back (TP9), as well as from the Cz location, as depicted in Figure 2B. The ground electrode was placed in front of the right tragus, and the reference electrode was placed on the right mastoid.

Continuous electromyography (EMG) was recorded using the same OpenBCI Cyton Biosensing board at 250 Hz using three electrodes (Skintact (Leonhard Lang GmbH, Innsbruck, Austria)^®^ ECG self-adhesive electrodes); ground, reference, and EMG channel located on the extensor muscles in the right forearm spaced approximately 1 cm apart.

### 2.3. Experimental Setup

The experiment followed a counter-balanced within-subject design, divided into four blocks per experimental condition (tongue or hand movement), each containing 25 movement trials, with experimental condition movement changing after every 50th trial (see Figure 2D). Before the first tongue block, subjects received instructions that they should move the tip of their tongue from the rest position to touch their upper palate behind their teeth and keep the position for two seconds before returning to the rest position. The movement should be ballistic and with minimal force to avoid fatigue. One subject reported that the target position was the tongue’s rest position, but in this case, the instruction was to move the tongue backward on the palate instead. Before the first hand block, subjects were instructed on how to perform fast wrist extensions. The wrist extension was ballistic with minimal force. The wrist extension was then kept for two seconds before returning to the rest position. While performing the tongue and hand movements, the subjects were asked to sit as still as possible to avoid movement artifacts, blinking, and swallowing during the preparation and movement phase (see Figure 2C). Subjects performed a single test trial before each block, reminding them of the trial phases and the target movement, but this test trial was omitted in the analysis. Three 3-min long idle activity blocks were recorded too to ensure that there were enough data to extract the same number of epochs as for the movement classes [31] (see Figure 2D), and before each of them, subjects were instructed to sit relaxed with their eyes open.

Each movement trial consisted of three phases, which were presented to the subject in an interface shown on a screen (see Figure 2C). In the Preparation phase (2 s), the subjects were asked to stop moving, blinking and swallowing. In the Movement phase (2 s), the subjects were asked to perform the movement of the block (hand or tongue) at the cue (beginning of the phase) and hold it for the duration of the phase, while keeping all other movements to a minimum. In the Relax phase (4 s), subjects were able to move and readjust as they needed to. The current progress of the trial was shown in the interface as a horizontally moving vertical line, but the subjects were instructed not to follow it with their eyes, instead focusing their gaze on the yellow circle on the screen.

### 2.4. Signal Analysis

#### 2.4.1. Pre-Processing

Initially, the EEG was filtered using a zero-phase fifth-order Butterworth band-pass filter between 0.1 and 30 Hz, and a notch filter with quality 30 at 50 Hz was applied. The passband was selected such that features from the MRCP, and event-related desynchronization and synchronization could be included in the classification analyses [31]. The EMG was filtered using a zero-phase sixth-order Butterworth band-pass filter (20–100 Hz) and then again using a 50 Hz notch filter of quality 30 before enveloping the EMG using the real part of a Hilbert transform.

#### 2.4.2. EMG Onset Detection

To synchronize the wrist extensions with the continuous EEG, EMG was used. The onset of the movement was estimated and used for the synchronization. A movement onset was detected when the derivative of the enveloped EMG first crossed a manually set block threshold in the time range (−1 s; 1 s) around the movement cue (see Figure 3). If no movement onset was detected around the cue, the cue itself became the movement onset. This was the case for eight repetitions for participant 5 and one repetition for participant 7. All identified movement onsets were visually inspected.

#### 2.4.3. Epoch Extraction

The epochs were extracted with two different timings with respect to the movement/cue onset. Users trying to operate BCIs for control purposes can tolerate some delay after attempting to activate the BCI, which allows the system to use additional discriminative information that potentially can improve the BCI performance [32]. Epochs for control tasks were extracted in the range of <−2 s; 1 s> relative to the movement onset or cue onset of the movement phase (t=0—Figure 2C) for hand and tongue movements. As the induction of neuroplasticity requires immediate feedback after the movement onset [33], the epochs for rehabilitation purposes have to end by the movement onset. The exact timing of when the afferent somatosensory feedback should reach the cortex is not known although it is expected to not exceed 200–300 ms [33,34]. Therefore, the epochs were extracted from <−2 s; 0 s> relative to movement onset. Moreover, for inducing neuroplasticity the motor cortical activation needs to be specific for the rehabilitation purpose, hence only hand movements are included in this analysis and not tongue movements. Idle epochs were extracted for control and rehabilitation tasks using random start points from the idle recordings, at least one second apart, so some of them possibly overlapped. Extracted epochs that in any channel went outside the ±100 μV range were rejected from further analysis to avoid classifying artifacts.

### 2.5. Movement Detection

Features were extracted from the epochs, and offline movement detection was estimated by classifying each movement class (hand, tongue) against the idle activity epochs in the three use-case scenarios (using tongue for control, using hand for control, and using hand for rehabilitation). Additionally, classification using around-ear EEG channels was compared to classification using the Cz channel in order to assess the loss of detection accuracy by abandoning the traditional site for recording movement-related brain activity.

#### 2.5.1. Feature Extraction

From the epochs, two feature types were extracted from each channel: Temporal, and spectral features. Temporal features were built using the average of non-overlapping half-second intervals in each channel. Also, the channel-wise cross-correlation at t=0 was calculated between a template and each epoch. The template was subject-specific and calculated as the mean value across all movement epochs from the training dataset. Spectral features consisted of average power, computed using Welch’s method, in δ (0.5–4 Hz), θ (4–8 Hz), α (8–13 Hz) and β (13–30 Hz) bands in each channel.

#### 2.5.2. Classification

The features were classified using Random Forest (RF) with 128 trees, a linear Support Vector Machine (SVM), Linear Discriminant Analysis (LDA), and a K-Nearest Neighbours (KNN) algorithm with K being the square root of the number of samples in the training set.

The RF is an ensemble classifier that creates several decision trees based on a subset of the available features, and it works well on small training sets [35]. The LDA and SVM are two linear classifiers that are popular within BCI applications due to their low computational requirements and generalization properties. The LDA and SVM may not work well on non-linear data; therefore, KNN was tested as well since it is a non-linear classifier [35]. The classification problem was movement vs. idle epochs. The classification was performed for each subject separately, and the same number of movement and idle epochs was used in the classification. The classification was run three times: (1) using all available features, (2) only using the spectral features, and (3) only using the temporal features. The reported classification accuracy is the median of five-fold cross-validation results for each subject. The same folds were used for the feature and classifier comparisons. Random chance classifier accuracy was determined by the upper limit of 95% confidence interval (α=0.05) of a balanced 2-class classifier (p=0.5) for the median number of test epochs per subject (n=100) [36].

### 2.6. Statistical Analysis

To investigate the effect of the BCI purpose, electrode locations, feature types, and classifiers, a four-way repeated measures ANOVA was performed. The factors were: Purpose (three levels: Hand Rehabilitation, Hand Control, and Tongue Control), electrode location (two levels: Around-ear EEG, and Cz), feature type (three levels: All, temporal, and spectral), and classifier (four levels: RF, SVM, KNN, and LDA). A significant test was followed up using a posthoc test with Bonferroni’s correction to avoid multiple comparisons. A Greenhouse–Geisser correction was applied if the assumption of sphericity was violated. In all tests, statistical significance was assumed when *p* < 0.05.

## 3. Results

On average 1.3±2.8 hand epochs and 0.1±0.3 tongue epochs were rejected from further analysis. The results are summarized in Figure 4 and Figure 5 and Table 2. Based on the grand averages across subjects in Figure 4 no clear MRCPs can be seen in the channels around the ear, contrary to the Cz channel where an increase in negativity towards the movement onset is observed. The point of maximal negativity reaches a relative difference of approximately 10 μV with respect to the beginning of the epoch.

The movement detection performance varied depending on the combination of classifier (RF, SVM, KNN, and LDA), feature type (temporal, spectral, and the two combined), movement type (hand and tongue) and BCI purpose (rehabilitation and control), but all accuracies were in the range of 53–83%.

As summarized in Figure 5 and Table 2 it can be seen that the highest classification accuracies are generally obtained when using all features compared to spectral and temporal features alone. However, the statistical analysis revealed no significant effect of feature type on the classification accuracies (F_(1.1,10.1)_ = 4.5; p=0.06). The performance was very similar when using RF, SVM, and LDA, but the performance decreased significantly for KNN compared to the three other classifiers (F_(3,27)_ = 46.6; p<0.01). The classification accuracies obtained for control purposes were approximately 5–20 percentage points higher compared to those obtained for rehabilitation purposes with tongue movements having higher classification accuracies compared to hand movements. The difference in classification accuracies between the control scenarios and rehabilitation scenarios was statistically significant (F_(1.3,11.5)_ = 6.2; p=0.02). The median classification accuracies were significantly above the chance level for the vast majority of the combinations, and those that were below the chance level were primarily associated with classification performed with KNN. It should be noted that there is a considerable variation in the classification accuracies across participants. The classification accuracies obtained from around-ear EEG channels were similar to those obtained from Cz (F_(1,9)_ = 0.1; p=0.83). There was a significant interaction between electrode location and purpose (F_(2,18)_ = 6.7; p=0.01) where Cz was associated with higher classification accuracies in the two scenarios involving the hand (rehabilitation and control purpose), while around-ear EEG was associated with the highest accuracies for the control purpose using the tongue. Moreover, there was a significant interaction between the electrode location and feature type (F_(1.2,10.9)_ = 11.9; p<0.01) where Cz was associated with higher classification accuracies when using temporal features, and around-ear EEG was associated with the highest accuracies when using spectral features.

## 4. Discussion

In this study, it was shown that movement-related brain activity could be detected with accuracies significantly higher than chance level. The highest accuracies were associated with epochs containing information one second after the movement onset (for BCI control purposes), and tongue movements had higher classification accuracies compared to hand movements.

The highest median classification accuracies for the three scenarios using around ear-EEG reached 70%, 73%, and 83% for hand (rehabilitation purpose), hand (control purpose), and tongue (control purpose), respectively. These classification results were higher than what has been reported previously where close to chance level accuracies were obtained [24,25]. It should be noted though that this is based on a limited sample, and that there are differences in the methodology. Namely, Schalk et al. [24] used three ear electrodes (same as this study) with similar positioning around the ear but only used a single channel at a time for hand movement detection. In the current study, all around-ear channels were used at the same time, which might have added enough information to improve the detection accuracy above the chance level. On the contrary, ten around-ear EEG channels did not lead to higher classification accuracies [25]. Jochumsen et al. [25] used the combined information of ten ear-EEG channels with a similar classifier to the one used in this study (Random Forest with manual temporal, spectral and template features) and still did not detect hand movement significantly above chance level (60 %), while the RF classifier accuracy in this study was significantly higher than chance level (73%). The difference might be explained by the authors only using MRCP information (0.05–10Hz) [25], while this study included more information from the EEG spectrum, specifically the mu (8–13 Hz) and beta rhythm (13–30 Hz) which contain event-related desynchronization/synchronization. Indeed, the differentiating information might lie in higher frequency bands, as this study, as well as the previous ear-EEG studies on movement-related brain activity decoding report a lack of clear MRCP morphology in ear-EEG [25,37]. This is also supported by the classification accuracies obtained for around-ear EEG using spectral features which were higher than those obtained using temporal features in the current study.

Tongue movement has not been previously detected from around-ear EEG, but Kæseler et al. [28] have reported visible MRCP morphology on T7 and T8, but they have not collected channels closer to the ear, nor tried to detect tongue movement from ear-EEG only. In their work, the authors detected any of four different tongue movements with 94% accuracy using 64 scalp electrodes in an offline scenario with an RF classifier. In this study, a similar RF classifier can, in an offline scenario, detect a single type of tongue movement with 83% accuracy using only three electrodes around the ear. Generally, for BCI control purposes based on motor imagery, imagined hand movements have been utilized since EEG caps were used where electrodes from C3-4 and Cz were available. If around-ear EEG is used for recording motor cortical activity for control purposes, it should be considered using tongue movements instead since the tongue has a large cortical representation that is closer to the around-ear EEG electrodes than the cortical representation of the hand. Specific patient groups may benefit from using the movement-related brain activity from the tongue instead of the hands. On the contrary to hand function, the tongue function may be preserved in individuals with tetraplegia after, e.g., spinal cord injury or to some degree in patients with amyotrophic lateral sclerosis with limited bulbar symptoms. In these cases, the representation of the tongue in the primary motor cortex may be preserved and hence control signals for a BCI can be extracted.

This study attempted to investigate the practical concerns of a BCI that is operated using movement-related brain activity, but a number of concerns and limitations still remain unaddressed. One of the practical concerns is the use of concealed ear-EEGs. While the electrodes in the study were kept mostly outside hair, the electrodes were manually attached to the subjects and were connected via cables to an amplifier lying on a table, thereby not being concealed. For an inconspicuous and comfortable use, EEG electrodes and the amplifier would need to be contained and concealed in a device like headphones. However, the electrode setup itself of around-ear EEG has previously been rated high in terms of aesthetics and comfort by stroke patients, therapists and relatives [19]. The electrodes used in the study are wet self-adhesives, a currently non-viable option for use in a headphone-like device with fast equipping and removal. The development of novel devices with easily attachable and detachable self-adhesives is, therefore, needed. Alternatively, the validation of the classification results on dry electrodes is needed for real-world use, as dry electrodes have higher impedance and using them can result in lower classification accuracy [38]. Additionally, dry electrodes can cause discomfort [38], which should be evaluated against the convenience of the dry setup. An alternative could be to use water-based electrodes that have been rated much higher in comfort compared to dry electrodes [19].

Another practical limitation is testing on able-bodied subjects that executed the movement. BCI use for both control and rehabilitation tasks involves individuals who have motor impairments. The detection accuracy of the classifiers in this study has possibly been improved by information from motor artifacts such as glossokinetic potentials occurring after the movement onset in tongue trials (see Figure 4). Epochs of motor-impaired patients may not contain the movement artifacts related to the attempted movement (although they may contain artifacts pertaining to unrelated movements) and the detection accuracy would, therefore, be affected. Additionally, when performing motor imagery, spinal-cord injury subjects have been found to reach lower MRCP and Event-related Desynchronisation/Synchronisation amplitudes [39] and reach lower detection accuracy than able-bodied individuals [40]. However, no significant detection accuracy differences have been found between stroke patients and able-bodied people [41]. Moreover, the BCI users would operate the BCI by attempting to perform the movement although no movement may occur on the contrary to performing motor imagery where the execution is voluntarily inhibited. There is a need to test around-ear EEG for decoding movement-related brain activity with the intended BCI users.

Also, BCIs can only be practically used in an online scenario, but currently, all studies including this one, perform offline analysis only. Offline analysis suffers from inflated accuracies, as results are reported on discrete epochs of either target movement or no movement. It is expected that the BCI performance will decrease, but it is not known by how much. It has been reported that a true positive rate of 67% is sufficient for inducing plasticity [12], which is lower than the 70% that was obtained in the current study. Moreover, it is expected that the 83% obtained in the current study, and possibly lower accuracies, can be used to control external technology by, e.g., using a binary brain switch based on MRCP detection to select different control commands in a cyclic menu that has been shown in able-bodied participants [42]. It has previously been shown that completely paralyzed patients can control a spelling device with a similar detection performance [7], and that amyotrophic lateral sclerosis patients could control an assistive active glove [5]. In patients with amyotrophic lateral sclerosis [5] and stroke [43,44] it has been reported that a similar BCI performance is obtained when using MRCPs as control signals. Despite the possibility of converting modest classification accuracies into relevant control commands using state machines, the BCI control will be slow, i.e., the information transfer rate will be low. This will be a bottleneck for adopting the technology by the end-user. Therefore, possibilities for improving the information transfer rate should be explored. For control purposes, steady-state visual evoked potentials can be recorded using ear-EEGs [45], or the motor cortical activity can be paired with the evoked potentials in a hybrid approach to increase the number of classes or improve the classification accuracies [46], which ultimately will improve the information transfer rate. Another approach for improving the classification accuracies could be to use additional signal processing techniques for pre-processing such as a blind source separation if enough electrodes are available [47] or investigate other classification techniques. Data-driven approaches such as deep learning could potentially improve the performance if there are enough data to train the classifier which could be obtained after multiple sessions of BCI use, or a pre-existing network could be used and adapted to the individual user.

## 5. Conclusions

This study has demonstrated the possibility of detecting tongue and hand movements solely from around-ear EEG using a low-cost amplifier. The best performance was obtained when using information from two seconds prior to one second after the movement onset. Tongue movements were associated with the best performance. Additional research is required to develop a practical ear-EEG BCI concealment and to validate the results in an online scenario with the intended BCI user group.

## Figures and Tables

**Figure 2 sensors-24-06004-f002:**
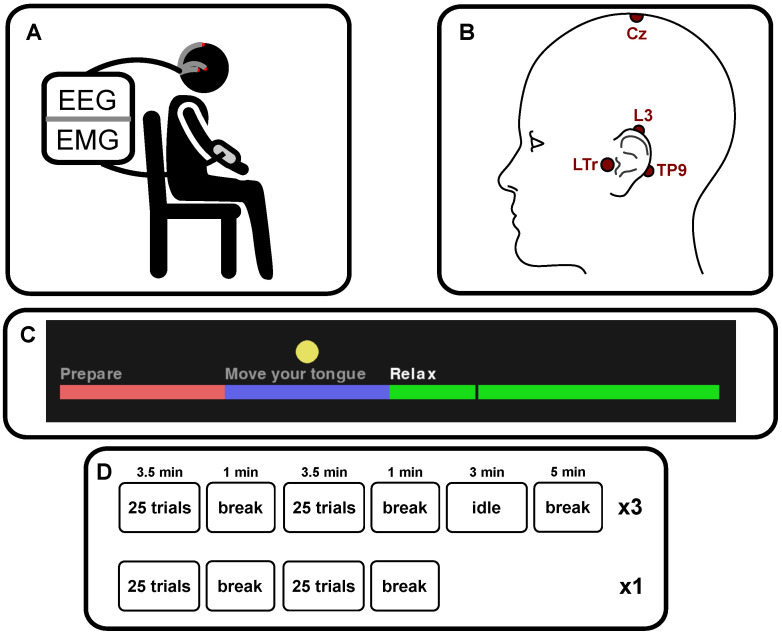
Visualization of aspects of the study’s experimental procedure. (**A**) depicts electroencephalography (EEG) and electromyography (EMG) signal collection from a sitting subject. (**B**) shows electrode locations for EEG data collection; LTr, L3, TP9 and Cz, also referred to as Front, Over, Back ear and scalp EEG. The ground and reference were around the right ear. (**C**) is a screenshot from the interface shown to subjects during cued movements. The vertical dark bar moved continuously from left to right during each trial, showing the trial’s progress. Participants were told to focus their gaze on the yellow circle and initiate the movement immediately when the cursor entered the blue area. (**D**) visualizes the experiment’s blocks and the breaks between them.

**Figure 3 sensors-24-06004-f003:**
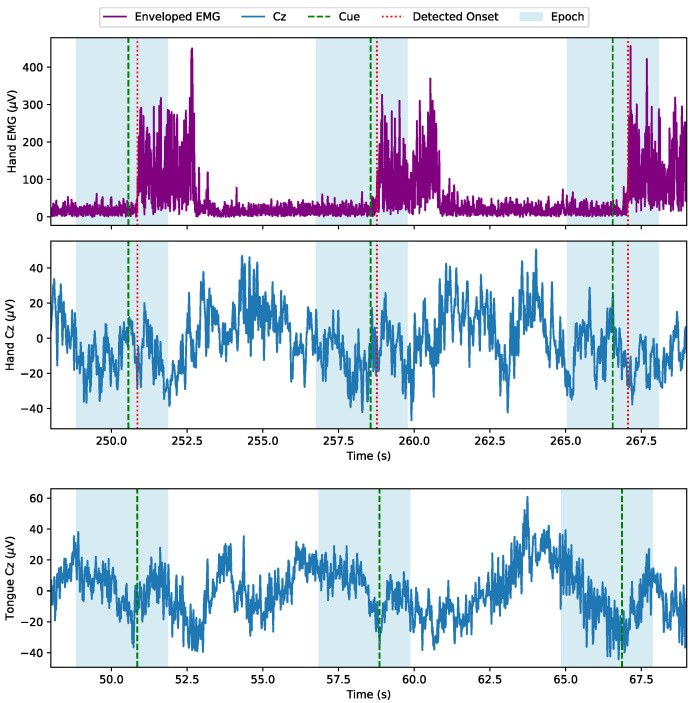
Epoch extraction visualized on three example epochs. Hand epochs are extracted using the EMG recorded from wrist extension (**top panel**). The hand movements are not timed precisely at the cue and the figure shows that onset detection is robust against the delay variation. The corresponding single-trial EEG is shown on the middle panel. Tongue epochs are extracted solely from the timing of the cues (**bottom panel**).

**Figure 4 sensors-24-06004-f004:**
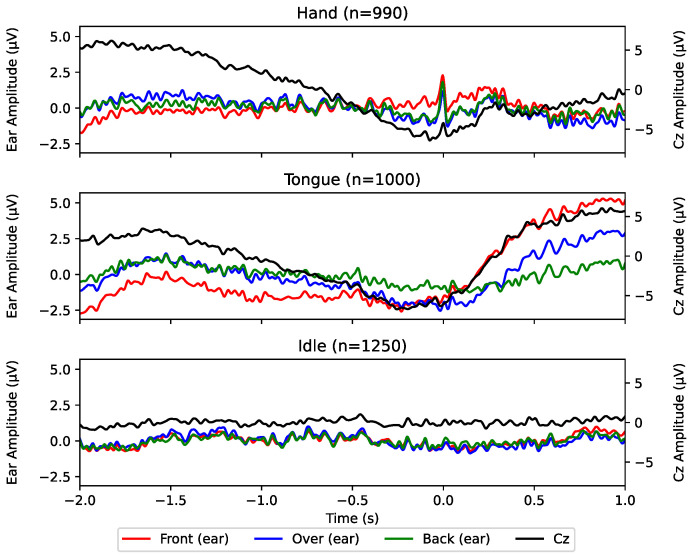
Grand epoch averages for each channel and condition across all subjects, centered (t=0) around movement onset (hand) or cue (tongue). *n* designates the number of averaged epochs. Ear-EEG channels and Cz channels are scaled differently in order to showcase the morphology more clearly. The vertical dotted lines denote the timing of negative peaks for each channel, labeled in seconds relative to the movement onset (t=0).

**Figure 5 sensors-24-06004-f005:**
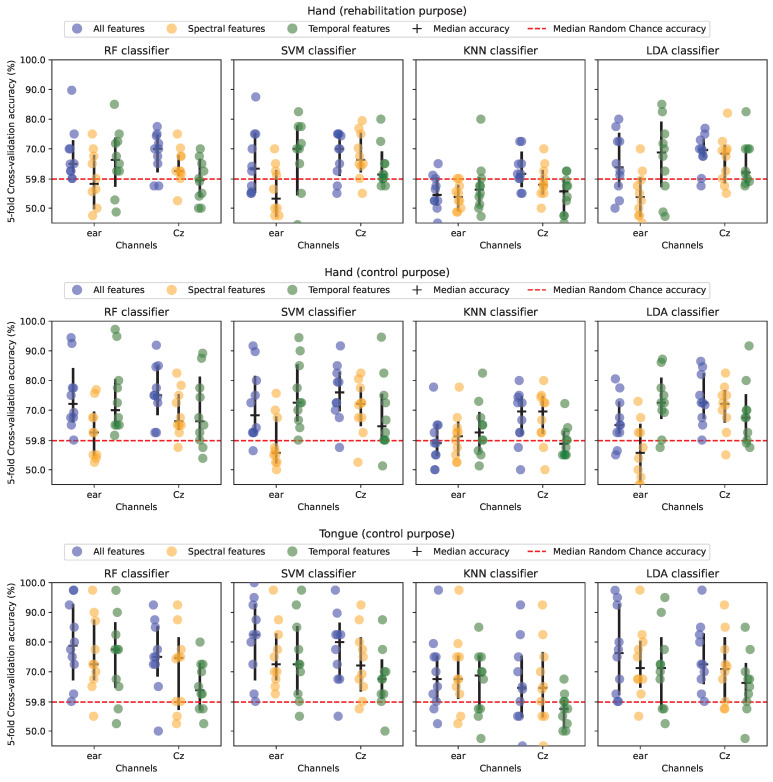
Median classification accuracy for each of the subjects in each of the scenarios and feature subsets. Each point represents a subject’s median classification accuracy of a five-fold cross-validation. Horizontal segments represent median accuracy across subjects and error bars depict a non-parametric 95% confidence range. RF: Random Forest. SVM: Support Vector Machine. KNN: K-Nearest Neighbours. LDA: Linear Discriminant Analysis.

**Table 2 sensors-24-06004-t002:** Median classification accuracy across subjects when using around-ear channels and Cz. RF: Random Forest. SVM: Support Vector Machine. KNN: K-nearest Neighbours. LDA: Linear Discriminant Analysis.

Condition	Classifier	Feature Type	Median Ear (%)	Median Cz (%)
Hand (Rehabilitation purpose)	RF	All	64.93	70.00
		Temporal	58.21	62.50
		Spectral	66.25	59.49
	SVM	All	63.30	70.00
		Temporal	53.21	66.25
		Spectral	70.00	61.32
	KNN	All	54.46	61.54
		Temporal	53.75	57.92
		Spectral	56.25	55.67
	LDA	All	63.75	69.62
		Temporal	53.75	68.37
		Spectral	68.75	62.02
Hand (Control purpose)	RF	All	72.12	75.00
		Temporal	62.50	66.25
		Spectral	70.00	66.25
	SVM	All	68.30	75.99
		Temporal	55.71	72.15
		Spectral	72.50	64.58
	KNN	All	58.97	69.58
		Temporal	61.25	69.58
		Spectral	62.50	58.75
	LDA	All	65.00	72.50
		Temporal	55.67	72.15
		Spectral	72.50	67.50
Tongue (Control purpose)	RF	All	78.75	75.00
		Temporal	72.50	74.68
		Spectral	77.50	63.75
	SVM	All	82.50	80.00
		Temporal	72.50	72.12
		Spectral	72.50	67.50
	KNN	All	67.50	64.55
		Temporal	67.50	64.55
		Spectral	68.75	57.50
	LDA	All	76.25	72.50
		Temporal	71.25	70.90
		Spectral	71.25	66.25

## Data Availability

The data presented in this study are available upon reasonable request from the authors in anonymized form due to GDPR.

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
