# Peer review of "Detection of Movement-Related Brain Activity Associated with Hand and Tongue Movements from Single-Trial Around-Ear EEG"

_sensors, 2024, doi:10.3390/s24186004_

Round 1

Reviewer 1 Report

Comments and Suggestions for Authors

Summary. This research addresses a significant challenge in BCI research, which is the development of more practical, comfortable, affordable, and less visible BCI systems. This study uses healthy controls to detect movement intentions from the hand/tongue for control and rehabilitation purposes. The results of this study suggest that on ear-EEG may be a potential solution to this long-standing challenge. Additionally, the high classification accuracies suggest that ear-EEG can be used for both rehabilitation purposes and to control external devices. This work has relevance to individuals with motor impairments (e.g., ALS).

General Comments. Overall, the article provides a thorough background of the potential of BCIs for individuals with motor impairments and the challenges associated with the current scalp EEG-based systems. Moreover, the authors have clearly motivated the need for more practical, comfortable, and aesthetically pleasing BCI solutions for patients.

There are a few areas where the introduction could be improved. Generally, readability and flow should be addressed. Particularly, sentence structure and cohesion. For example, smoother paragraph transitions to guide the reader though the logical progression of ideas. Some areas of the article should be more concise. It will make the manuscript clearer and readable to a wider audience. More specific comments are provided below.

SPECIFIC COMMENTS

INTRODUCTION

Line 19: The clinical importance of this research can be bolstered by adding references of how quality of life is impaired in individuals with disorders.

 Lines 18-26: Consider replacing “severe” with “debilitating” on line 22 to better describe conditions like ALS. Savić et al., 2021 included relatively mild participants. I suspect as ALS progresses to a clinically severe stage, conventional pathways likely become inaccessible and more challenging to extract brain activity/control BCI. Is there research that can be referenced on individuals with late-stage ALS?

Line 27: Consider starting a new paragraph for readability. I would also consider switching paragraphs 2 and 3 in the introduction. Paragraph 3 seems to clearly identify limitations in the field and motivates the study.

Line 39: This may be more appropriate in the methods section.

Line 99: Combine this with the last paragraph of the introduction.

I was able to see the Tables, however, I did not see any Figures embedded in the manuscript, just the Figure descriptions.

METHODOLOGY

Line 147: What was the rationale for the 3-minute duration of the idle activity blocks? Was this duration based on previous literature?

How was the timing of the cue for movement onset determined? Was it continuous based on the timing of the phases?

Lines 161-165: Please provide references or the rationale for the pre-processing procedures.

Line 171: The manuscript mentions that if no movement was detected, the cue itself became the movement onset. How often did this occur in this dataset? Is it common?

Line 187: What was the rationale the ±100µV amplitude threshold?

RESULTS

Include statistical tests to determine the significance of the observed differences in classification accuracies. This will help strengthen the discussion section.

DISCUSSION

It was briefly mentioned that “it is expected that the 83% obtained in the current study, and possibly lower accuracies, can be used to control external technology…” Is this in healthy controls? Please elaborate on the potential clinical implications of the findings, particularly for individuals with motor impairments.

Line 256: “affected” and “the same potential issue” are vague. Please be more precise.

The discussion could also benefit from more discussion on the tongue. The spectral information is comparable to the temporal. Using both temporal and spectral together achieved a higher accuracy for tongue initiation. Would tongue information be more useful for certain populations/situations.

DATA AVAILABILITY/ETHICS STATEMENTS

All information is provided to show that the study was carried out with ethical research standards.

REFERENCES

The cited references are mostly recent publications and relevant. 

Reviewer 2 Report

Comments and Suggestions for Authors

The article is certainly devoted to the interesting topic of recognizing human activity by its brain activity. The authors emphasize the use of more affordable EEG devices with a small number of electrodes. The big question for this article is the complete absence of graphic materials that are not inserted into the text. I have not found additional materials, so it is extremely difficult to give a final assessment of this work without seeing the results and graphical visualization of the author's approach. The second question is why the authors limit themselves only to models of a random forest and a support vector machine, although it is not so difficult to consider 2-3 more algorithms to show the effectiveness of the chosen approaches. Thus, these two comments are serious in the sense that it is difficult to give an objective and complete assessment of the article without eliminating them.

Round 2

Reviewer 2 Report

Comments and Suggestions for Authors

It should be noted that correcting those significant comments that were noted earlier allows us to assess the level and quality of the article. The work is definitely relevant and promising in technical, scientific and practical terms. The authors present a fairly voluminous graphic material, thanks to which the methodology and results of the study are understandable.

What comments do I have left after reading the revised version:

1 The authors have expanded the set of machine learning methods they use. It is wonderful. I would like to see a small paragraph in section 2.5.2, where it will be indicated why these methods were chosen (ease of implementation, reliability, interpretability, etc.), and why the rest were not used or will be considered in the future.

2 In the conclusion or discussion section, I would like to see 1 paragraph devoted to the practical application of the results obtained, since for a number of experiments the accuracy is still low (60-70%). I would like to see a comment on how this can interfere with the application and what further steps will be taken in the next studies.

Thus, my assessment of the article is positive, after eliminating these minor comments, the article can be published.
